# COMPOSITIONAL CONSERVATISM: A TRANSDUCTIVE APPROACH IN OFFLINE REINFORCEMENT LEARNING

**Yeda Song**[∗]**, Dongwook Lee**[∗] **& Gunhee Kim**
Seoul National University
yeda.song@vision.snu.ac.kr, {dwsmart32, gunhee}@snu.ac.kr

## ABSTRACT

Offline reinforcement learning (RL) is a compelling framework for learning optimal policies from past experiences without additional interaction with the environment. Nevertheless, offline RL inevitably faces the problem of distributional shifts, where the states and actions encountered during policy execution may not be in the training dataset distribution. A common solution involves incorporating conservatism into the policy or the value function to safeguard against uncertainties and unknowns. In this work, we focus on achieving the same objectives of conservatism but from a different perspective. We propose COmpositional COnservatism with Anchor-seeking (COCOA) for offline RL, an approach that pursues conservatism in a *compositional* manner on top of the transductive reparameterization (Netanyahu et al., 2023), which decomposes the input variable (the state in our case) into an anchor and its difference from the original input. Our COCOA seeks both in-distribution anchors and differences by utilizing the learned reverse dynamics model, encouraging conservatism in the compositional input space for the policy or value function. Such compositional conservatism is independent of and agnostic to the prevalent *behavioral* conservatism in offline RL. We apply COCOA to four state-of-the-art offline RL algorithms and evaluate them on the D4RL benchmark, where COCOA generally improves the performance of each algorithm. The code is available at https://github.com/runamu/compositional-conservatism.

## 1 INTRODUCTION

Reinforcement learning (RL) has achieved notable successes across various domains, from guiding robotic movements (Dasari et al., 2020) and optimizing game strategies (Mnih et al., 2015) to recently promising training of language models (Rajpurkar et al., 2016). Despite these achievements, the challenges posed by real-time interaction in complex and sensitive environments have prompted the development of offline RL as a viable direction. Offline RL (Wiering & Van Otterlo, 2012; Levine et al., 2020) or batch RL (Lange et al., 2012) learns policies solely from pre-existing data, without any direct interaction with the environment. Offline RL is becoming increasingly popular in real-world applications such as autonomous driving (Yu et al., 2020a) or healthcare (Gottesman et al., 2019) where prior data are abundant.

By its nature of learning from prior datasets, offline RL is often susceptible to distributional shifts. This issue arises when the distribution of states and actions encountered during policy execution differs from that of the training dataset, a situation particularly challenging in machine learning (Levine et al., 2020). Numerous existing offline RL algorithms tackle this by reducing distributional shifts through conservative approaches, including constraining the policy or estimating uncertainty to measure distributional deviations (Kim & Oh, 2023; Ran et al., 2023; Kostrikov et al., 2022; Kumar et al., 2020; Wu et al., 2019; Kumar et al., 2019; Fujimoto et al., 2019; Sun et al., 2023; Rigter et al., 2022; Wang et al., 2021; Yu et al., 2021; 2020b; Kidambi et al., 2020). These strategies aim to keep the agent within known distributions, mitigating risks of unexpected behaviors. In this work, we also pursue the same goal of conservatism, focusing on aligning the test data distribution with the seen distribution, but from a different perspective.

---

[∗]Authors equally contributed.

We begin by recognizing that the state distributional shift problem is closely related to addressing how to deal with the out-of-support input points of the function approximators. We explore the possibility of transforming the out-of-support learning problem into an out-of-combination problem by injecting inductive biases into function approximators of the policy or the Q-value function. Such a transformation has been previously proposed by Netanyahu et al. (2023), where a transductive approach named *bilinear transduction* makes predictions through a bilinear architecture after reparameterizing the target function. This reparameterization decomposes the input variable into two components, namely an *anchor* and a *delta*, where the anchor is a variable in the input space and the delta is the difference between the input variable and the anchor. If the reparameterized training and test data distribution satisfy certain assumptions, and if the target function has certain properties, the bilinear transduction can address the out-of-combination problem, which potentially resolves the out-of-support problem with the original target function.

We propose COmpositional COnservatism with Anchor-seeking (*COCOA*) for offline RL, a framework that adopts a compositional approach to conservatism, building upon the transductive reparameterization (Netanyahu et al., 2023). Our approach transforms the distributional shift problem into an out-of-combination problem. This shifts the key factors for generalizability from data to decomposed components and the interrelations between them, demanding the anchor and delta to be selected close to the training dataset distribution.

We suggest a new anchor-seeking approach with an additional policy, named *anchor-seeking policy*, which enforces the agent to find anchors within the seen area of the state space. Hence, COCOA encourages anchors to be close to the offline dataset while confining the deltas within a narrow range by identifying anchors among neighboring states. This approach can reduce the input space and guide it toward the space predominantly explored during the training phase. In summary, by learning a policy to seek in-distribution anchors and differences from the learned dynamics, we can encourage conservatism in the compositional input space of the function approximator for the Q-function and policy. This approach is independent of and agnostic to the prevalent behavioral conservatism in offline RL.

We empirically find that our method improves the performance of four representative offline RL methods, including CQL (Kumar et al., 2020), IQL (Kostrikov et al., 2022), MOPO (Yu et al., 2020b), and MOBILE (Sun et al., 2023) on the D4RL benchmark (Fu et al., 2020). We also show through an ablation study that learning anchor-seeking policy is effective in improving the performance of our method. Our main contributions can be summarized as follows:

- We pursue conservatism in the *compositional input space* for the function approximators of the Q-function and policy, independently and agnostically to the prevalent *behavioral* conservatism in offline RL.

- We introduce COmpositional COnservatism with Anchor-seeking (*COCOA*) that finds in-distribution anchors and deltas with the learned dynamics model, which is crucial for compositional generalization.

- We empirically show that COCOA improves the performance of four state-of-the-art offline RL algorithms on the D4RL benchmark. Additionally, our ablation study shows the efficacy of anchor-seeking policy compared to heuristic anchor selection.

## 2 PRELIMINARIES

### 2.1 OFFLINE RL

We assume an MDP problem $(\mathcal{S}, \mathcal{A}, T, R)$ with a continuous state space $\mathcal{S}$, a continuous action space $\mathcal{A}$, a transition function $T : \mathcal{S} \times \mathcal{A} \rightarrow \mathcal{S}$, and a reward function $R : \mathcal{S} \times \mathcal{A} \rightarrow \mathbb{R}$. The goal is to find a policy $\pi : \mathcal{S} \rightarrow \mathcal{A}$ that maximizes the expected return $J(\pi) = \mathbb{E}_\pi \left[ \sum_{t=0}^{\infty} \gamma^t R\left(s_t, a_t\right) \right]$, where $\gamma \in [0, 1)$ is a discount factor.

In offline RL, also known as batch RL, we are given a dataset $\mathcal{D}_{\text{env}} = \{(s_i, a_i, s_{i+1}, r_i)\}_{i=1}^{N}$ generated with a behavior policy. The goal in offline RL is to find a policy $\pi$ that maximizes the expected return $J(\pi)$ using only the fixed dataset $\mathcal{D}_{\text{env}}$. Like most model-baed offline RL algorithms, we learn a dynamics model $\widehat{T}(s_{i+1}|s_i, a_i)$ that predicts a next state $s_{i+1}$ given a current state $s_i$ and action $a_i$.

In addition to the forward dynamics model, we also learn a reverse dynamics model $\widehat{T}(s_i|s_{i+1}, a_i)$ that predicts a current state $s_i$ given a next state $s_{i+1}$ and action $a_i$.

## 2.2 Bilinear Transduction

We follow the formulation of Netanyahu et al. (2023) about the generalization problem. With no assumption on the train and test distribution, the generalization performance of a function approximator is limited. This problem occurs especially when the test distribution is not contained in the train distribution, also known as an out-of-support (OOS) learning problem. As a special case of OOS, an out-of-combination (OOC) problem occurs when the input space is decomposed into two components, and the marginal of the train distribution of each component includes that of the test distribution while the joint train distribution does not necessarily contain the joint test distribution. Under certain assumptions, Netanyahu et al. (2023) propose a transductive reparameterization method called *bilinear transduction* to convert an OOS problem into an OOC problem.

**Bilinear transduction.** It solves extrapolation under certain assumptions. First, the target function $f(x)$ is reparameterized as

$$f(x) := \bar{f}(x - \tilde{x}, \tilde{x}), \tag{1}$$

where $\tilde{x}$ is termed as an *anchor*, selected from the training dataset. The difference $(x - \tilde{x})$ between the input variable $x$ and the anchor $\tilde{x}$ is termed as a *delta*. The reparameterized target function $\bar{f}$ is approximated as a bilinear function of the embeddings $\boldsymbol{\varphi_1}$ and $\boldsymbol{\varphi_2}$:

$$\bar{f}_{\boldsymbol{\theta}}(x) = \boldsymbol{\varphi_1}(x - \tilde{x}) \cdot \boldsymbol{\varphi_2}(\tilde{x}). \tag{2}$$

Intuitively, it facilitates the low-rank property of the embeddings $\boldsymbol{\varphi_1}$ and $\boldsymbol{\varphi_2}$, enabling the function approximator to generalize to OOC points.

**Sufficient conditions for bilinear transduction.** Netanyahu et al. (2023) introduce sufficient conditions for bilinear transduction to be applicable. The assumptions are about both the dataset and the target function $f$. The first assumption is about a *combinatorial coverage* of the dataset. The test dataset has to have a bounded combinatorial density ratio with respect to the training dataset. It implies that the support of the joint distribution of the training distributions of the components should include the support of the joint distribution of the test distributions of the components. Second, the target function $f$ should be *bilinearly transducible*, i.e., there exists a deterministic function $\bar{f}$ such that $f(x) = \bar{f}(x - \tilde{x}, \tilde{x})$ for all $x, \tilde{x} \in \mathcal{X}$. Lastly, the training distribution of anchors should not degenerate (Shah et al., 2020). Under these three conditions, it is possible to generalize the target function to OOC points with a theoretically guaranteed risk bound.

**Connection to compositional generalization.** In light of the literature on *compositional generalization* (Wiedemer et al., 2023), we interpret bilinear transduction as a special case of compositional generalization, where the $\boldsymbol{\varphi_1}, \boldsymbol{\varphi_2}$ models serves as *component functions*, extracting low-rank features of the input, and the inner product serves as a *composition function*.

# 3 Compositional Conservatism with Anchor-seeking (COCOA)

## 3.1 Offline RL with Bilinear Transduction

The base algorithms in offline RL, such as Deep Q-Networks (DQN) (Mnih et al., 2015) and Actor-Critic methods (Mnih et al., 2016; Haarnoja et al., 2018), frequently use deep neural networks as function approximators. Hence, we employ bilinear transduction (§ 2.2) to the function approximators of policy and Q-function. In both train and test phases, we decompose the current state $s$ into an anchor $\tilde{s}$ and a delta $\Delta s = s - \tilde{s}$, where $\tilde{s} \sim \mathcal{D}_{\text{env}}$. Then, the policy and the Q-function will be

$$\bar{\pi}_{\boldsymbol{\theta}}(s) = \boldsymbol{\varphi_{\theta,1}}(\Delta s) \cdot \boldsymbol{\varphi_{\theta,2}}(\tilde{s}), \quad \bar{Q}_{\boldsymbol{\phi}}(s, a) = \boldsymbol{\varphi_{\phi,1}}(\Delta s, a) \cdot \boldsymbol{\varphi_{\phi,2}}(\tilde{s}, a). \tag{3}$$

The architecture of the policy and Q-function is illustrated in Figure 1c. The policy $\pi(a|s)$ is trained to maximize the expected return $J(\pi)$, and the Q-function $Q(s, a)$ is trained to minimize its loss function $\mathcal{L}_Q$ defined in the base offline RL algorithm.

Different state decompositions can result in different compositional input spaces, resulting in different generalization capabilities. In order to satisfy the assumptions of bilinear transduction in § 2.2,

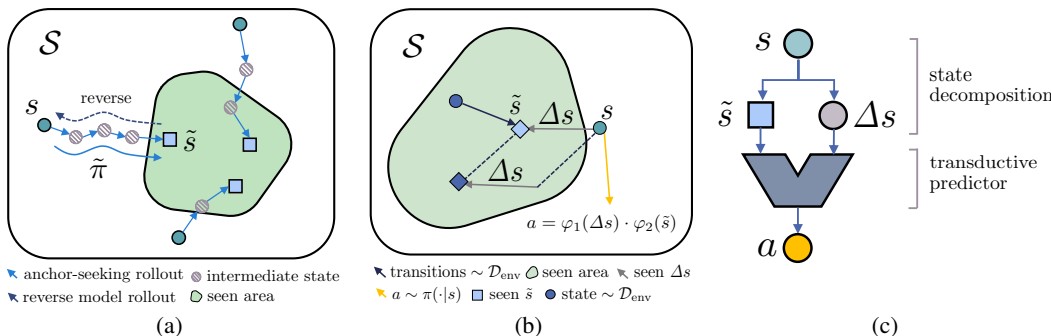

Figure 1: **(a)** An illustration of anchor-seeking rollouts that find anchors close to the seen area of the state space $\mathcal{S}$. Given the current state $s$, the anchor-seeking policy $\tilde{\pi}$ gives actions to reach the anchor $\tilde{s}$. Its behavior is derived by utilizing reverse model rollouts, which diverge from the offline dataset. **(b)** An illustration of the current state $s$ and an anchor $\tilde{s}$. Ideally, the anchor $\tilde{s}$ has been observed during the training phase when it served as an anchor for another state. Similarly, the difference, delta, had also been encountered previously in the ideal case but in combination with a different anchor. **(c)** The architecture of our policy $\pi(a|s)$ that aims to generalize to an unfamiliar state by decomposing the state $s$ into familiar components (seen anchor $\tilde{s}$ and seen delta $\Delta s$) and applying a transductive predictor. The architecture of the Q-function is similar to that of the policy.

the ideal approach is to find the decomposition that fulfills these two criteria: in-distribution anchor and in-distribution delta. This ideal case is illustrated in Figure 1c. However, in contrast to the prior works (Netanyahu et al., 2023; Pinneri et al., 2023) that only focus on the transducible property of the goal state, we try to handle each state in every step, and it is computationally infeasible to enforce these constraints using brute-force methods like comparing the current states to all other points. Hence, we introduce a new anchor-seeking policy that learns to find in-distribution anchors and deltas and exploit the learned dynamics model that prevents arbitrary decomposition, to exploit the power of bilinear transduction. We also enforce each delta to be within a distance of a few steps of the dynamics model to confine the distribution of delta in both the train and test phase to a similar range. This approach reduces the input space and guides it toward the space that is predominantly explored during the training phase, thereby further enhancing generalizability.

## 3.2 LEARNING TO SEEK IN-DISTRIBUTION DECOMPOSITION

We describe the reverse dynamics model, the anchor-seeking trajectory, and the random divergent reverse policy, which are required components before training the anchor-seeking policy.

### 3.2.1 ANCHOR-SEEKING TRAJECTORY

**Training a reverse dynamics model.** Given a transition $(s, a, s')$ sampled from the dataset $\mathcal{D}_{\text{env}}$, we train a reverse transition dynamics model $\widehat{T}_r(s|s', a)$ (Wang et al., 2021; Lai et al., 2020; Goyal et al., 2018; Edwards et al., 2018; Holyoak & Simon, 1999) to predict the state $s$ given the next state $s'$ and action $a$. In other words, the reverse dynamics model $T(s', a)$ predicts "From which state $s$ do we come if we arrive at $s'$ by taking action $a$?". This is done by minimizing the loss function with the dataset's state $s$ defined as

$$\mathcal{L}_{\text{r}} = \mathbb{E}_{(s,a,s')\sim\mathcal{D}_{\text{env}}} \left[ \left\| \widehat{T}_r(s', a) - s \right\|_2^2 \right].  \tag{4}$$

**Random divergent reverse policy.** We do not use a trained reverse policy but instead use a heuristic reverse policy that randomly selects an action from the dataset $\mathcal{D}_{\text{env}}$. The subsequent actions in reverse rollouts, after the initial action, follow the same direction as the initial action but are slightly scaled down with a small added Gaussian noise. This ensures that the reverse rollout diverges away from the dataset. Since we use random actions and maintain a consistent direction throughout the reverse rollout, it is more likely to venture into unexplored regions beyond the offline dataset. In

---

**Algorithm 1** Generation of Anchor-Seeking Trajectory

---

**Require:** Offline dataset $\mathcal{D}_{\text{env}}$, reverse dynamics $\widehat{T}_r$, anchor-seeking horizon $h$, rollout epoch $e$.
1: **for** $k$ in $1 \ldots e$ **do**
2:     Sample anchor state $s_t \sim \mathcal{D}_{\text{env}}$.
3:     Generate reverse model rollout $\widehat{\tau} = \{(s_{t-i}, a_{t-i}, r_{t-i}, s_{t+1-i})\}_{i=1}^{h}$ from $s_t$ by using the reverse dynamics $\widehat{T}_r$ and random divergent actions $\{a_{t-i}\}_{i=1}^{h}$.
4:     Add model rollouts to replay buffer, $\mathcal{D}_{\text{reverse}} \leftarrow \mathcal{D}_{\text{reverse}} \cup \{(s_{t-i}, a_{t-i}, r_{t-i}, s_{t+1-i})\}_{i=1}^{h}$.
5: **end for**
6: **return** $\mathcal{D}_{\text{reverse}}$

---

**Algorithm 2** Anchor-Seeking Rollout

---

**Require:** State $s$, anchor-seeking policy $\tilde{\pi}$, forward dynamics model $\widehat{T}$, anchor-seeking horizon $h$.
1: Set the initial state for anchor-seeking: $\tilde{s} = s$.
2: **for** $i = 1$ to $h$ **do**
3:     Compute the anchor-seeking action $\eta \leftarrow \tilde{\pi}(\tilde{s})$.
4:     Update anchor $\tilde{s} \leftarrow \widehat{T}(\tilde{s}, \eta)$.
5: **end for**
6: **return** $\tilde{s}$

---

summary, the reverse policy gives an action $a_j$ at each rollout step $j$ as follows:

$$a_j = \phi a + \epsilon_j, \quad \text{where} \quad a \sim \mathcal{D}_{\text{env}}, \epsilon_j \sim \mathcal{N}(0, \sigma^2), \quad j = 1, 2, \ldots, h. \tag{5}$$

$h$ is a horizon length, $\phi$ is a scale coefficient, and $\sigma$ is a noise coefficient. We set $\phi = 0.8$ and $\sigma = 0.1$ when the maximum action value is 1.0.

**Anchor-seeking trajectory.** We use rollouts of the reverse model to make anchor-seeking trajectories for training the anchor-seeking policy. First, we sample an anchor state from the dataset and generate a reverse transition $\mathcal{D}_{\text{reverse}} = \{(s_{i+1}, a_i, s_i, r_i)\}_{i=1}^{j}$ from the anchor state using the reverse dynamics model and the random divergent reverse policy. Note that the direction of anchor-seeking trajectory is reverse to that of reverse transition, $\mathcal{D}_{\text{reverse}}$. By doing so, we can effectively generate anchor-seeking trajectories for training of the anchor-seeking policy. Utilizing reverse model rollouts to address the OOD problem is first proposed by Wang et al. (2021), who augment the offline dataset with reverse transition, train a policy using this augmented dataset, and demonstrate the efficacy of such an approach in the offline RL setting. Finally, the detail of generating anchor-seeking trajectory is summarized in Algorithm 1.

### 3.2.2   TRAINING OF DYNAMICS-AWARE ANCHOR-SEEKING POLICY

We train the anchor-seeking policy $\tilde{\pi}(a|s)$ before training the main policy. We utilize anchor-seeking trajectories in § 3.2.1, which are the reverse direction to the dataset $D_{\text{reverse}}$. By following the path of an anchor-seeking trajectory, the anchor-seeking policy is trained to select actions $\eta$ that guide the agent in a direction moving from the external boundary towards the seen area as illustrated in Figure 1a. Given the anchor-seeking rollout is generated by the anchor-seeking policy $\tilde{\pi}(a|s)$ and the dynamics model $\widehat{T}(s, a)$, the reverse transition $D_{\text{reverse}}$ relies on $\widehat{T}_r(s, r|s', a)$. Since the reverse transition was designed to diverge from the offline dataset, the anchor-seeking trajectory, with its reversed direction, ensures the transition converges back to the dataset from unfamiliar states. As a result, we train the anchor-seeking policy to minimize the MSE loss between the predicted action and the action in the dataset $D_{\text{reverse}}$. The loss function is defined as

$$\mathcal{L}_{\text{anchor}}(\theta) = \mathbb{E}_{(s', a, s) \sim \mathcal{D}_{\text{reverse}}, \eta \sim \tilde{\pi}_\theta(a|s)} \left[ (\eta - a)^2 \right]. \tag{6}$$

In this way, the anchor-seeking policy $\tilde{\pi}(a|s)$ can provide a proper action to move toward an in-distribution anchor. This action is given to the dynamics model $\widehat{T}(s, a)$ to predict the next state. Thereby, as shown in Figure 1, the anchor-seeking rollout is a sequence of transitions that starts from the current state $s$ and ends at the anchor state $\tilde{s}$.

---

**Algorithm 3** Bilinear Transduction with Anchor-Seeking (Actor)

---

**Require:** Input state $s$, anchor-seeking policy $\tilde{\pi}$, forward dynamics model $\widehat{T}$, anchor-seeking horizon $h$, embedding layers $\{\boldsymbol{\varphi_1}, \boldsymbol{\varphi_2}\}$.
1: Start from $s_0 \leftarrow s$ and perform anchor-seeking rollout $\widehat{\tau} = \{(s_i, a_i, s_{i+1})\}_{i=0}^{h-1}$ by using the anchor-seeking policy $\tilde{\pi}(a_i|s_i)$ and forward dynamics $\widehat{T}(s_{i+1}|s_i, a_i)$.
2: Decompose the state $s$ by using the final state as an anchor: $\tilde{s} \leftarrow s_h$, $\Delta s \leftarrow s - s_h$.
3: Bilinear transduction of the target function: $\bar{f} \leftarrow \boldsymbol{\varphi_1}(s - \tilde{s}) \cdot \boldsymbol{\varphi_2}(\tilde{s})$.
4: Process $\bar{f}$ with an additional MLP layer: $\bar{f} \leftarrow \mathrm{MLP}(\bar{f})$.
5: **return** $\bar{f}$

---

### 3.3 SUMMARY OF THE METHOD

We illustrate the integration of the anchor-seeking model into the bilinear transduction framework. We choose the Soft Actor-Critic (SAC) algorithm (Haarnoja et al., 2018) as a representative RL algorithm that employs function approximators. We utilize the bilinear transduction supplemented with anchor-seeking for the SAC's actor and critic networks.

Given an input state $s_n$, we use rollout to obtain an action from the anchor-seeking policy. The anchor $\tilde{s}$ is derived by this action through the forward step of dynamics and then updated to be the next state $s_{n+1}$. After rolling out a few times we can pinpoint the final anchor and use it to decompose the state into the anchor and delta. The difference between the initial state $s_n$ and this anchor $\tilde{s}$, delta, is computed as $\Delta s = \tilde{s} - s_n$.

We then carry out the bilinear transduction as described in Eq.(2). We respectively embed $\Delta s$ and $\tilde{s}$ as $\boldsymbol{\varphi_1}(\Delta s)$ and $\boldsymbol{\varphi_2}(\tilde{s})$, and compute the inner product between them. The output is then fed into a small MLP layer to enhance the flexibility of the function approximator. This step introduces nonlinearity, as the policy or Q-function may not be linear to their inputs.

Algorithm 3 summarizes the whole process in the actor module. In the critic module, we concatenate the action with both the anchor and delta in the forward operation before executing bilinear transduction. Subsequently, the values of action and Q-value, derived as $\bar{f}_{\mathrm{actor}}$ and $\bar{f}_{\mathrm{critic}}$ respectively, are employed to update the actor and critic networks of the SAC policy.

## 4 EXPERIMENTS

In our experiments, we aim to empirically answer the following two questions: (i) how much does our method improve the performance of prior model-free and model-based algorithms? and (ii) what is the effect of the anchor-seeking on performance?

### 4.1 RESULTS OF D4RL BENCHMARK TASKS

We evaluate our method on the Gym-MuJoCo tasks in D4RL benchmark (Fu et al., 2020), which consists of 12 tasks from the OpenAI Gym (Brockman et al., 2016) and MuJoCo (Todorov et al., 2012) environments. Refer to A.1 for the details of the tasks.

**Baselines.** We apply COCOA to several prior offline RL algorithms, both model-based and model-free methods. They include (i) CQL (Kumar et al., 2020) that penalizes Q-values on out-of-distribution samples for safety, (ii) IQL (Kostrikov et al., 2022) that leverages the generalization capability of the function approximator by viewing the state value function as a random variable, (iii) MOPO (Yu et al., 2020b) as a model-based approach that penalizes rewards based on uncertainty from predicting subsequent states, and (iv) MOBILE (Sun et al., 2023) that quantifies uncertainty through the inconsistency of the Bellman estimation using an ensemble of dynamics models. We also report the results of (v) Behavior Cloning (BC), which learns tasks by imitating expert data. To stabilize training, all baseline algorithms are reproduced with layer normalization applied.

**Results.** Table 1 summarizes the results of our experiments. The baseline algorithms are denoted as "Alone", and our method is denoted as "+COCOA". We report the average return of the last 10 training epochs across 4 seeds, with the standard deviation. For all algorithms, we reproduce the

Table 1: Results of the D4RL benchmark. We report the normalized average return of the last 10 training epochs across 4 seeds on the D4RL benchmark tasks.

| Task | BC | CQL | | IQL | | MOPO | | MOBILE | |
|---|---|---|---|---|---|---|---|---|---|
| | | Alone | +COCOA | Alone | +COCOA | Alone | +COCOA | Alone | +COCOA |
| halfcheetah-random | 2.2 | 31.3 | 23.8 ± 0.8 | - | - | 37.3 | 33.9 ± 2.3 | 40.9 | 29.5 ± 2.4 |
| hopper-random | 3.7 | 5.3 | **8.8 ± 1.6** | - | - | 31.7 | **32.3 ± 1.2** | 17.4 | 11.5 ± 8.3 |
| walker2d-random | 1.3 | 5.4 | **5.5 ± 8.5** | - | - | 4.1 | **23.7 ± 0.5** | 9.9 | **21.4 ± 0.1** |
| halfcheetah-medium | 43.2 | 46.9 | **49.0 ± 0.2** | 47.4 | 47.1 ± 0.1 | 72.4 | 70.8 ± 3.7 | 73.5 | 69.2 ± 1.0 |
| hopper-medium | 54.1 | 61.9 | **66.0 ± 1.4** | 66.3 | 65.5 ± 1.5 | 62.8 | 39.5 ± 2.3 | 92.9 | **103.7 ± 6.1** |
| walker2d-medium | 70.9 | 79.5 | **83.1 ± 0.3** | 78.3 | **81.9 ± 0.2** | 84.1 | **89.4 ± 1.4** | 80.8 | **84.4 ± 1.3** |
| halfcheetah-medium-replay | 37.6 | 45.3 | **46.4 ± 0.4** | 44.2 | 40.3 ± 1.4 | 72.1 | 67.0 ± 0.5 | 66.8 | **68.6 ± 0.9** |
| hopper-medium-replay | 16.6 | 86.3 | **96.5 ± 1.9** | 94.7 | 88.2 ± 5.9 | 92.8 | 71.4 ± 30.1 | 87.9 | **105.3 ± 1.7** |
| walker2d-medium-replay | 20.3 | 76.8 | **83.9 ± 2.9** | 73.9 | 66.9 ± 10.0 | 85.2 | **93.1 ± 5.1** | 81.1 | **83.5 ± 1.7** |
| halfcheetah-medium-expert | 44.0 | 95.0 | 92.4 ± 3.1 | 86.7 | **91.4 ± 1.4** | 83.6 | **109.1 ± 1.4** | 86.75 | **104.1 ± 2.0** |
| hopper-medium-expert | 53.9 | 96.9 | **103.3 ± 1.4** | 91.5 | **105.3 ± 2.0** | 74.9 | **101.4 ± 13.9** | 102.3 | **109.1 ± 4.7** |
| walker2d-medium-expert | 90.1 | 109.1 | **109.4 ± 0.4** | 109.6 | 109.1 ± 0.9 | 105.3 | **109.1 ± 0.4** | 107.0 | **107.5 ± 0.9** |
| Average | 36.5 | 61.6 | **64.0** | 77.0 | **77.3** | 67.2 | **70.1** | 70.6 | **74.8** |

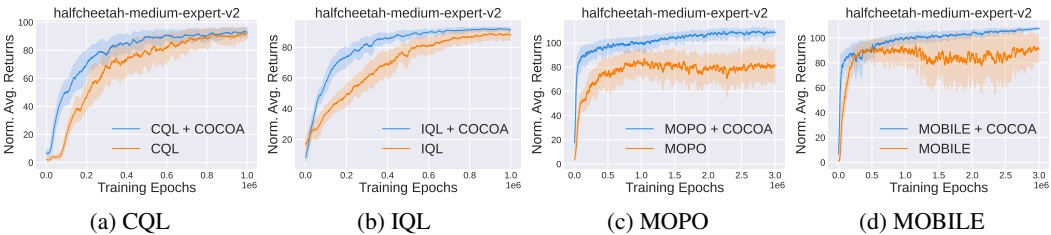

(a) CQL      (b) IQL      (c) MOPO      (d) MOBILE

Figure 2: Performance of CQL, IQL, MOPO, and MOBILE with and without COCOA on the halfcheetah-medium-expert-v2 task of D4RL.

results with the codebase described in Appendix A.3. Our method enhances the performance of all baseline algorithms, as measured by the average improvement across tasks excluding the random tasks of IQL as the original IQL paper does. In summary, our COCOA improves the performance of the original methods in 10 out of 12 tasks for CQL, 3 out of 9 tasks for IQL, 7 out of 12 tasks for MOPO, and 9 out of 12 tasks for MOBILE.

## 4.2 ABLATION STUDY: THE EFFECT OF ANCHOR-SEEKING

To examine the impact of anchor selection on performance, we experiment with a variant of our method that does not use anchor-seeking. For this ablation study, we use CQL (Kumar et al., 2020) as the base algorithm due to its computational efficiency as a model-free algorithm and its completeness in supporting all task types, including "random" tasks.

**Baseline.** The baseline for this ablation study is denoted as "+COCOA (w/o A.S.)" in Table 2. In this baseline, we adapt the heuristic anchor selection procedure from Netanyahu et al. (2023), incorporating key modifications for our context. Unlike the original method, which selects anchors based on goal states, our baseline selects anchors based on the current state, addressing the absence of a goal state in our setting. To mitigate the computational demands of this method, we limit our selection to a subset of candidate anchors, randomly sampled from the dataset.

The heuristic anchor selection operates as follows. We initially draw $N$ candidate anchors $s_i$ from the dataset and calculate the difference $\Delta s$ between the candidates and the current state, defined by

$$\Delta s_n = s - s_n, \quad n \in \{1, \dots, N\}, \quad s_n \in D_{\text{env}}. \tag{7}$$

Subsequently, we assess every pairwise difference among another $N$ sampled states from $D_{\text{env}}$ as

$$\Delta s_{i,j} = s_i - s_j, \quad i, j \in \{1, \dots, N\}, \quad i \neq j, \quad s_i, s_j \in D_{\text{env}}. \tag{8}$$

Finally, we select the candidate anchor that minimizes the distance to the current state:

$$\tilde{s} = s_{\tilde{n}}, \quad \text{with} \quad \tilde{n} = \arg\min_{n} \left\{ \min_{i,j} \| \Delta s_n - \Delta s_{i,j} \| \right\}. \tag{9}$$

This baseline enforces the results of state decomposition to be close to in-distribution data through direct distance calculation. While it can be effective and feasible if the dataset is small and $N$ is sufficiently large, its scalability is limited as the required amount of computation scales quadratically with the data size. Since the computation cost escalates cubically with the sample size, we set $N$ to 30, matching our computation budget with "+COCOA".

**Results.** We examine whether this variant improves the performance of CQL. The results are summarized in Table 2. We report the average return of the last 10 training epochs across 4 seeds, with the standard deviation. The baseline "+COCOA (w/o A.S.)" achieves higher performance in only two tasks, "hopper-random" and "walker2d-random", and comparable or lower performance in the other tasks compared to the original "Alone" baseline. In contrast, our method "+COCOA" improves the performance of the CQL models in 10 out of 12 tasks. This result suggests that the anchor-seeking is a crucial component for the success of our method.

Table 2: Ablation study for anchor-seeking. We report the normalized average return of the last 10 training epochs across 4 seeds on the D4RL benchmark tasks.

| Task | CQL | | |
|---|---|---|---|
| | Alone | +COCOA (w/o A.S.) | +COCOA |
| halfcheetah-random | **31.3** | 22.5 ± 0.5 | 23.8 ± 0.8 |
| hopper-random | 5.3 | **25.8 ± 7.4** | 8.8 ± 1.6 |
| walker2d-random | 5.4 | **8.7 ± 5.3** | 5.5 ± 8.5 |
| halfcheetah-medium | 46.9 | 47.6 ± 0.2 | **49.0 ± 0.2** |
| hopper-medium | 61.9 | 54.0 ± 2.4 | **66.0 ± 1.4** |
| walker2d-medium | 79.5 | 80.3 ± 0.9 | **83.1 ± 0.3** |
| halfcheetah-medium-replay | 45.3 | 45.1 ± 0.3 | **46.4 ± 0.4** |
| hopper-medium-replay | 86.3 | 84.7 ± 2.9 | **96.5 ± 1.9** |
| walker2d-medium-replay | 76.8 | 78.7 ± 2.2 | **83.9 ± 2.9** |
| halfcheetah-medium-expert | **95.0** | 15.8 ± 3.2 | 92.4 ± 3.1 |
| hopper-medium-expert | 96.9 | 30.4 ± 9.8 | **103.3 ± 1.4** |
| walker2d-medium-expert | 109.1 | 86.8 ± 2.3 | **109.4 ± 0.4** |
| Average | 61.6 | 48.4 | **64.0** |

## 5 RELATED WORK

**Offline RL.** In offline RL, agents use a predefined dataset without additional interactions with the environment, typically following either the model-based or model-free strategy. Model-free RL algorithms (Kim & Oh, 2023; Ran et al., 2023; Kostrikov et al., 2022; Kumar et al., 2020; Wu et al., 2019; Kumar et al., 2019; Fujimoto et al., 2019) optimize policy directly using prior experiences in the replay buffer, applying conservatism to the value function or policy. In contrast, model-based offline RL methods (Sun et al., 2023; Rigter et al., 2022; Wang et al., 2021; Yu et al., 2021; 2020b; Kidambi et al., 2020) use a model trained in the environment to create additional data that are employed for policy learning. Through such synthesized data, this approach becomes stronger in generalization and robust even to unseen states.

**Out-of-Distribution Generalization in Offline RL.** Much study has been done to improve the out-of-distribution (OOD) generalization of offline RL algorithms. Lou et al. (2022) tackles the action distributional shift problem by introducing a mutual information-based approach to learn an action embedding model. In a similar pursuit, Gu et al. (2022) propose a pseudometric action representation learning method that measures both behavioral and data-distributional relations between actions. Bai et al. (2022) develop an uncertainty-driven method that uses the disagreement of bootstrapped Q-functions. It augments the dataset with OOD data on which a more refined penalty is imposed. Pitis et al. (2022) propose a local factorization of transition dynamics and state augmentation for improved generalization of offline RL algorithms. They also provide theoretical proofs for sample complexity and generalization ability. Our method is similar to theirs in that we also employ the factorized architecture of the policy and Q-function. Unlike them, however, we do not use a factorized dynamics model and instead leverage a bilinear transduction framework.

**Compositional Generalization and Extrapolation.** Compositional generalization, which strives to generalize to unseen combinations of components, is explored by various studies. Wiedemer et al. (2023) highlight a two-step generative procedure as essential for tackling a wide range of compositional problems. This procedure involves the complex generation of individual components and their straightforward combination into a single output. They provide a set of sufficient conditions under

which models trained on the data can generalize compositionally. On a related note, Shah et al. (2020) presents a sample-efficient RL algorithm that exploits the low-rank structure of the optimal Q-function, which is a bilinear function of states and actions. They prove a quantitative sample complexity improvement for RL with continuous state and action spaces via low-rank structure. Dong & Ma (2023) explore the extrapolation of nonlinear models for structured domain shift. They prove that a specific family of nonlinear models can successfully extrapolate to unseen distributions, provided the feature covariance is well-conditioned. (Netanyahu et al., 2023) propose an extrapolation strategy based on bilinear embeddings to enable combinatorial generalization, thereby addressing the out-of-support problem under certain conditions.

## 6 CONCLUSION

We explored a new perspective of conservatism for offline RL that does not regard the behavior space of the agent but the compositional input space of the policy and Q-function. We proposed a practical framework, COCOA, for finding better decomposition of states to encourage such conservatism. COCOA is a simple yet effective approach that can be applied to any offline RL algorithm that utilizes a function approximator. We empirically found that our method generally enhanced the performance of offline RL algorithms through our experiments across various tasks in the Gym-MuJoCo environment of the D4RL benchmark.

As our study primarily engages in empirical exploration, further investigation may be demanded for a more comprehensive understanding of the mechanism behind the performance improvement or the properties of the compositional input space. Moreover, since our experiments were limited to control-based robotics environments with continuous state and action spaces, it could be a valuable extension of this work to apply our compositional conservatism framework to other domains, including environments with discrete action spaces, image-based observations, or highly complex dynamics.

## 7 ACKNOWLEDGMENTS

We thank Jaekyeom Kim, Soochan Lee, Seohong Park, Aviv Netanyahu, and the anonymous reviewers for their valuable discussions and feedback. This work was supported by Institute of Information & Communications Technology Planning & Evaluation (IITP) grant funded by the Korean government (MSIT) (No. 2019-0-01082, SW StarLab), Institute of Information & communications Technology Planning & Evaluation (IITP) grant funded by the Korean government (MSIT) (No. 2022-0-00156, Fundamental research on continual meta-learning for quality enhancement of casual videos and their 3D metaverse transformation), Institute of Information & communications Technology Planning & Evaluation (IITP) grant funded by the Korean government(MSIT) [NO.2021-0-01343, Artificial Intelligence Graduate School Program (Seoul National University)], and Center for Applied Research in Artificial Intelligence(CARAI) grant funded by Defense Acquisition Program Administration(DAPA) and Agency for Defense Development(ADD) (UD190031RD). Gunhee Kim is the corresponding author.

## 8 REPRODUCIBILITY STATEMENT

For reproducibility, we provide the code of our method at https://github.com/runamu/compositional-conservatism. For the codebase for the baseline algorithms, please refer to Appendix A.3. The hyperparameters and the model architecture are described in Appendix A.4 and Appendix A.2, respectively.

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

## A   EXPERIMENT SETTINGS AND IMPLEMENTATION DETAILS

### A.1   D4RL BENCHMARK TASKS

**HalfCheetah**: The half-cheetah is a two-dimensional bipedal robot composed of 8 solid links, encompassing two legs and a torso, coupled with 6 motorized joints. The state space is 17-dimensional, encompassing both joint angles and velocities. An adversary destabilizes it by exerting a 6-dimensional action with 2-dimensional forces on the torso and each foot.

**Hopper**: The hopper is a planar monopod robot, assembled with 4 solid links that represent the torso, upper leg, lower leg, and foot, and includes 3 motorized joints. It has an 11-dimensional state space, including joint angles and velocities. An adversary employs a 2-dimensional force on the foot to disrupt its stability.

**Walker2D**: The walker operates as a two-dimensional bipedal robot with a structure of 7 links, representing two legs and a torso, along with 6 actuated joints. Within its 17-dimensional state space, joint angles and velocities are included. An adversary employs a 4-dimensional action with 2-dimensional forces on both feet to disrupt its equilibrium.

**Adroit**: Adroit is a complex task where a simulated robotic hand with 24 Degrees of Freedom is used for tasks such as hammering a nail, opening a door, spinning a pen, or moving a ball. We use two kinds of datasets for this: the "human" dataset, which includes 25 human demonstration trajectories, and the "cloned" dataset, which is an equal mix of demonstration data and behavior cloned from the demonstration policy.

**NeoRL**: NeoRL(Qin et al., 2022) is a benchmark designed to mirror real-world conditions by gathering datasets using a more cautious policy, aligning closely with realistic data collection methods. The scarcity and specificity of the data present a significant challenge for offline RL algorithms. Our research examines nine datasets, encompassing three different environments (HalfCheetah-v3, Hopper-v3, Walker2d-v3) and three levels of dataset quality (L, M, H), indicating low, medium, and high quality, respectively. Notably, NeoRL offers varying quantities of training data trajectories (100, 1000, 10000) for each environment. For our experiments, we uniformly chose 1000 trajectories.

### A.2   MODEL ARCHITECTURE

**Dynamics Model Architecture**: As with previous works, we used a neural network as the backbone for our dynamics model, which outputs a Gaussian distribution for the next state and reward. By ensembling these networks, we achieved greater stability and enhanced performance. From an ensemble of seven, we selected the top five models based on validation error. The backbone of the dynamics model comprises four layers, each with a hidden dimension of 200.

**Actor & Critic Architecture**: The actor-critic framework like SAC (Haarnoja et al., 2018) comprise actor and critic modules. Typically, an actor possesses a backbone constructed from a neural network. Features embedded within this backbone are relayed through a last layer that outputs a Gaussian distribution, yielding a non-deterministic result. Although MOPO, MOBILE, CQL, and IQL (Yu et al., 2020b; Sun et al., 2023; Kostrikov et al., 2022; Kumar et al., 2020), traditionally use 2, 2, 3, and 2 backbone layers with a dimension of 256 respectively, upon integrating COCOA, we standardized the use of two backbone layers with 100 hidden dimensions.

**Anchor-seeking Policy Architecture**: The anchor-seeking policy acts as an add-on module shared between the actor and critic. The input data, consisting of delta and anchor, is embedded through a neural network and subsequently processed by a bilinear architecture. Initially, inputs are embedded to the dimension of 4 with two neural networks with 64 channels, and the bilinear architecture produces an output with a dimension of 64 using those embedded features. Then, the outputs of bilinear architecture are passed through the actor and critic backbone architectures, resulting in the determination of the action and Q value, respectively.

**Parameter Size**: The anchor-seeking policy is built upon a compact neural network. For model-based algorithms like MOPO and MOBILE, the dynamics parameter size is approximately 1.9M, similar to that of COCOA. However, the parameter size needed to train the actor and critic for MOPO and MOBILE is equivalent to 0.21M. However, when COCOA is added to these algorithms,

the parameter size decreases to 0.19M. Given the significant magnitude of the dynamics parameters, the cumulative parameter requirement for training across COCOA-added model-based algorithms consistently stands at 2.2M. In contrast, IQL+COCOA and CQL+COCOA, which operate without a dynamics model, each have a parameter size of 2.0M.

## A.3 CODE IMPLEMENTATION

Our method is designed as an add-on enhancement to existing offline RL algorithms. Consequently, rather than developing a new implementation, we adapted the established codebases of base algorithms. For consistent and reliable code adaptation, we relied on Sun (2023) as the foundation for all base algorithms, including CQL (Kumar et al., 2020), IQL (Kostrikov et al., 2022), MOPO (Yu et al., 2020b) and MOBILE (Sun et al., 2023). The reliability of this codebase is supported by detailed training logs and results that align with those in the original papers. Additionally, Sun (2023) offers results for the Gym-MuJoCo-v2 datasets not present in the original CQL and MOPO papers, satisfying our needs. Note that an author of MOBILE (Sun et al., 2023) provides this codebase. Our adaptations to the code are shared as a demo in the supplementary material.

## A.4 HYPERPARAMETERS FOR EACH ALGORITHM

**CQL.** For both CQL and CQL+COCOA, we use $\alpha = 5.0$ for all D4RL-Gym tasks because the reproduced codebase (Sun, 2023) which provides the results for MuJoCo-v2 tasks, which are not included in the original paper (Kumar et al., 2020), uses this value. For COCOA, the anchor seeking horizon length $h$ was set to 1 for most tasks, except for "halfcheetah-medium-expert-v2", "hopper-medium-expert-v2", and "walker2d-medium-expert-v2", where $h$ was set to 3.

**IQL.** For both IQL, we use the same hyperparameters described in the original paper (Kostrikov et al., 2022), $\tau = 0.7$ and $\beta = 3.0$, which is also used in the reproduced codebase (Sun, 2023). For IQL+COCOA, we used $\tau = 0.6$ and $\beta = 3.0$. For COCOA, we set the anchor seeking horizon length $h$ to 1 for all tasks. We reproduce the random value for halfcheetah, hopper, walker2d, which are 6.62 to 6, 8.1 to 7, 6.1 to 6.5 respectively.

**MOPO.** For MOPO, we use the hyperparameters used in the reproduced codebase (Sun, 2023), which provides the results for MuJoCo-v2 tasks not included in the original paper (Yu et al., 2020b). As in the original paper, we use aleatoric uncertainty for MOPO and MOPO+COCOA. For MOPO+COCOA, we search for the best penalty coefficient $\lambda$ and rollout length $h_r$ for each task in the following ranges: $\lambda \in \{0.1, 0.5, 1.0, 5.0, 10.0\}$, $h_r \in \{1, 5, 7, 10\}$ except for the case of halfcheetah-medium-expert. The best hyperparameters are described in Table 3. For COCOA, we set the anchor seeking horizon length $h$ to 1 for all tasks.

**MOBILE.** We use the same hyperparameters described in the original paper (Sun et al., 2023) for MOBILE. For MOBILE+COCOA, we search for the best penalty coefficient $\lambda$ and rollout length $h_r$ for each task in the following ranges: $\lambda \in \{0.1, 1.0, 1.5, 2.0\}$, $h_r \in \{1, 5, 10\}$ except for the case of walker-medium-replay. The best hyperparameters are described in Table 3. For COCOA, we set the anchor seeking horizon length $h$ to 1 for all tasks. Additionally, after verifying convergence, we limited our training to a maximum of 2000 epochs and obtained results from this specific epoch range.

## A.5 ADDITIONAL EXPERIMENTAL RESULTS

We experimented with two more benchmarks - D4RL Adroit and NeoRL. The outcomes of these experiments are summarized in the table 4 and 5. This broader analysis reveals that COCOA enhances the performance of IQL and MOBILE in most tasks. All experiments on additional benchmarks were conducted without applying layer normalization to allow for direct comparison with the performance reported in their original papers.

Our method demonstrated consistent performance enhancements across six D4RL Adroit tasks we tested, showcasing its robustness and adaptability. While COCOA encountered challenges in complex tasks like the door and hammer, akin to its original algorithm, this reflects the inherent difficulty of these tasks due to sparse rewards. Notably, in tasks like the pen, our method achieved noticeable performance improvements.

Table 3: Hyperparameters for MOPO+COCOA and MOBILE+COCOA on D4RL benchmark.

| Task | MOPO+COCOA | | MOBILE+COCOA | |
|------|------------|------|--------------|------|
| | $\lambda$ | $h_r$ | $\lambda$ | $h_r$ |
| halfcheetah-random | 0.1 | 7 | 1.0 | 10 |
| hopper-random | 10.0 | 5 | 0.1 | 1 |
| walker2d-random | 0.5 | 5 | 1.0 | 5 |
| halfcheetah-medium | 0.5 | 7 | 1.0 | 5 |
| hopper-medium | 5.0 | 10 | 1.5 | 10 |
| walker2d-medium | 1.0 | 7 | 1.0 | 10 |
| halfcheetah-medium-replay | 1.0 | 10 | 1.0 | 10 |
| hopper-medium-replay | 0.1 | 5 | 1.0 | 10 |
| walker2d-medium-replay | 0.5 | 10 | 1.0 | 3 |
| halfcheetah-medium-expert | 2.5 | 5 | 2.5 | 10 |
| hopper-medium-expert | 5.0 | 10 | 2.0 | 10 |
| walker2d-medium-expert | 1.0 | 10 | 1.0 | 10 |

Table 4: Adroit benchmark results. We report the normalized average return of the last 10 training epochs across 2 seeds on the Adroit benchmark tasks.

| Task | IQL | | MOBILE | |
|------|------|--------|--------|--------|
| | Alone | +COCOA | Alone | +COCOA |
| door-cloned-v1 | 2.11 | **2.81** | -0.27 | **0.92** |
| door-human-v1 | 5.27 | **6.81** | -0.28 | **-0.05** |
| hammer-cloned-v1 | 0.53 | **0.57** | 0.23 | **0.29** |
| hammer-human-v1 | 1.33 | **2.21** | 0.25 | **1.11** |
| pen-cloned-v1 | 72.55 | **73.98** | 54.95 | 48.9 |
| pen-human-v1 | 74.4 | **78.15** | 14.76 | **41.65** |
| Average | 26.03 | **27.22** | 11.61 | **15.47** |

In Adroit, the training epoch is limited to 200 as described in (Sun et al., 2023). Plus, we used the same hyperparameters for MOBILE+COCOA on Adroit as described in the paper. Hyperparameters for Adroit and NeoRL benchmark are described in Table 6.

### A.6    COMPARISON WITH COMBO ALGORITHM

CQL+COCOA and COMBO exhibit some similarities, notably in their use of dynamics and a less conservative approach towards state-action space. However, their methodologies in pursuing conservatism differ significantly: COCOA focuses on conservatism in the compositional input space, whereas COMBO emphasizes regularizing the values for unfamiliar actions. Thus, COCOA and COMBO are orthogonal, it would be an insightful comparison to integrate COCOA with COMBO, as COCOA can be an add-on to any algorithm.

Similar to COMBO, MBPO-based methods like MOPO and RAMBO also show a tendency to outperform model-free methods in random and medium settings. It seems that data augmentation through MBPO is particularly beneficial in these tasks. It would be interesting to theoretically or empirically compare the state-specific value functions among CQL, CQL+COCOA, and COMBO for further analysis.

Table 5: NeoRL benchmark results. We report the normalized average return of the last 10 training epochs across 4 seeds on the NeoRL benchmark tasks.

| Task | MOPO | MOPO+COCOA |
|------|------|------------|
| HalfCheetah-L | 40.1 | **47.47** |
| Hopper-L | 6.2 | **23.02** |
| Walker2d-L | 11.6 | **14.23** |
| HalfCheetah-M | 62.3 | **78.26** |
| Hopper-M | 1 | **61.65** |
| Walker2d-M | 39.9 | **49.80** |
| HalfCheetah-H | 65.9 | 42.44 |
| Hopper-H | 11.5 | **29.31** |
| Walker2d-H | 18 | **53.38** |
| Average | 28.5 | **44.40** |

Table 6: Hyperparameters of MOBILE+COCOA for Adroit benchmark and MOPO+COCOA for NeoRL benchmark.

| Task | $\lambda$ | $h_r$ |
|------|-----------|-------|
| door-cloned-v1 | 0.5 | 7 |
| door-human-v1 | 3 | 3 |
| hammer-cloned-v1 | 3 | 1 |
| hammer-human-v1 | 5 | 1 |
| pen-cloned-v1 | 0.5 | 1 |
| pen-human-v1 | 10 | 1 |
| HalfCheetah-v3-low | 0.5 | 5 |
| Hopper-v3-low | 2.5 | 5 |
| Walker2d-v3-low | 2.5 | 1 |
| HalfCheetah-v3-medium | 0.5 | 5 |
| Hopper-v3-medium | 1.5 | 5 |
| Walker2d-v3-medium | 2.5 | 1 |
| HalfCheetah-v3-high | 1.5 | 5 |
| Hopper-v3-high | 2.5 | 5 |
| Walker2d-v3-high | 2.5 | 1 |

# B DETAILED RESULTS

## B.1 THEORETICAL ANALYSIS

Here, we simply show how our transductive predictor can be approximated in bilinear operation with the Stone-Weierstrass theorem.

**Theorem 1 (Stone-Weierstrass Theorem for Locally Compact Spaces)** *Suppose $X$ is a locally compact Hausdorff space and $A$ is a subalgebra of $C_0(X, \mathbb{R})$. Then $A$ is dense in $C_0(X, \mathbb{R})$ with respect to the topology of uniform convergence if and only if it separates points and vanishes nowhere.*

Let $\mathcal{X}$ and $\mathcal{Y}$ be locally compact Hausdorff (LCH) spaces. Plus, let $\mathcal{F} \subset C(\mathcal{X}; \mathbb{R})$ and $\mathcal{G} \subset C(\mathcal{Y}; \mathbb{R})$ be dense subvector spaces in the topology of uniform convergence on compacta. Then the Theorem 1 tells us that

$$\left\{ \sum_{k=1}^{d} f_k(x) g_k(y) \,\middle|\, f_1, \ldots, f_k \in \mathcal{F}, g_1, \ldots, g_k \in \mathcal{G}, d \in \mathbb{N} \right\} \subseteq C(\mathcal{X} \times \mathcal{Y}; \mathbb{R})$$

, which forms an algebra, is dense in the topology of uniform convergence on compacta. In other words, if we have a joint embedding $f_\theta \colon \mathcal{X} \to \mathbb{R}^d$ and $g_\phi \colon \mathcal{Y} \to \mathbb{R}^d$, then $h_{\theta,\phi}(x, y) = f_\theta(x) \cdot g_\phi(y)$

is a universal approximator, such that $(d, width) \to (\infty, \infty)$ and $f_\theta(x)$, $g_\phi(y)$ have depth $\geq 2$. Considering we utilize a parameterized network to approximate a transductive predictor and our input space $(s, a)$ is a subset of $\mathbb{R}^m \times \mathbb{R}^n$, where $m$ and $n$ denote their respective dimensions, $\boldsymbol{\varphi_{\theta, 1}}$ and $\boldsymbol{\varphi_{\theta, 2}}$ which are described in Section 3.1, can correspond to $f_\theta$ and $g_\phi$, respectively.

## B.2 PERFORMANCE GRAPHS OF D4RL BENCHMARK TASKS

In this section, we provide the performance graphs of each algorithm on D4RL benchmark tasks. We include only the 9 tasks that are not "random" tasks because the checkpoints of the baseline methods for the "random" tasks are not provided.

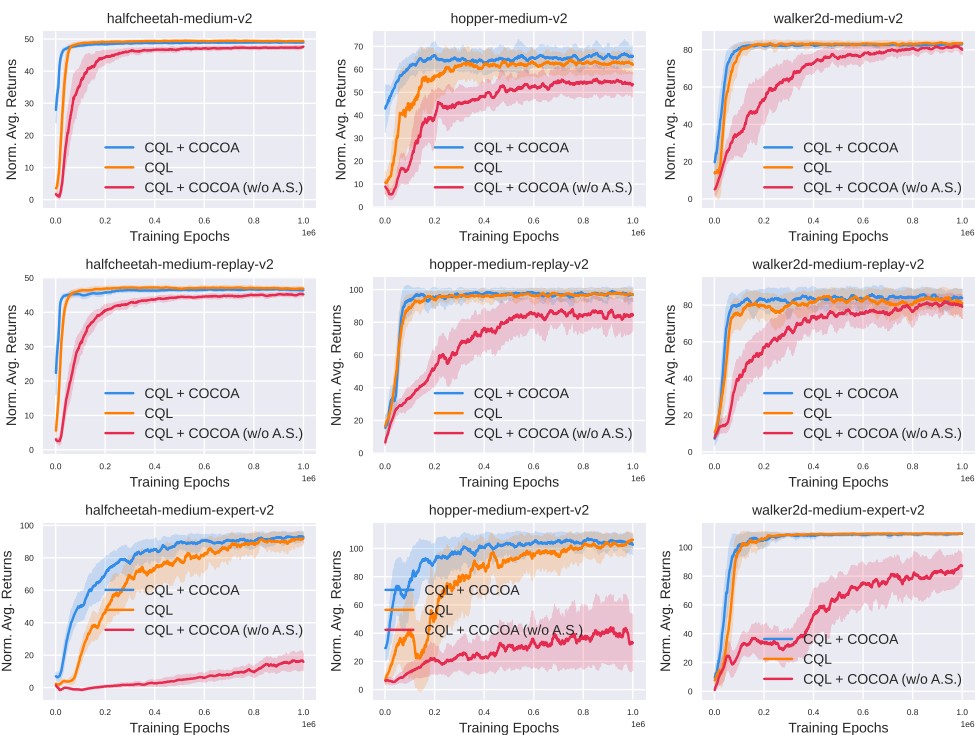

Figure 3: Performance comparison of CQL, CQL+COCOA and CQL+COCOA without anchor-seeking across all D4RL tasks except for "random" tasks.

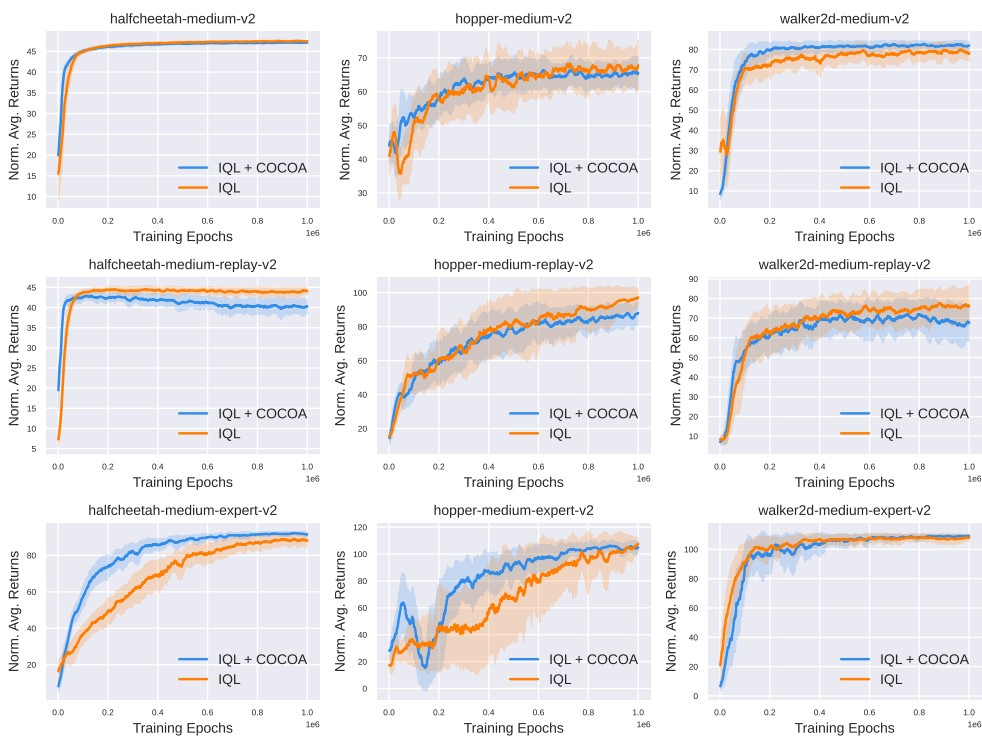

Figure 4: Performance comparison of IQL and IQL+COCOA across all D4RL tasks except for "random" tasks.

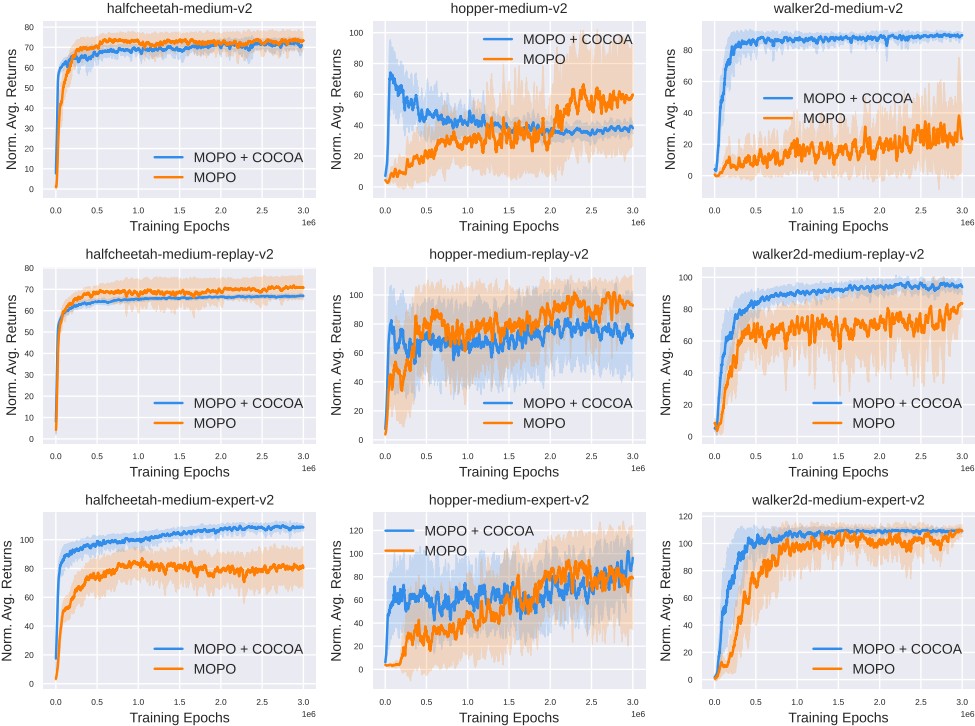

Figure 5: Performance comparison of MOPO and MOPO+COCOA across all D4RL tasks except for "random" tasks.

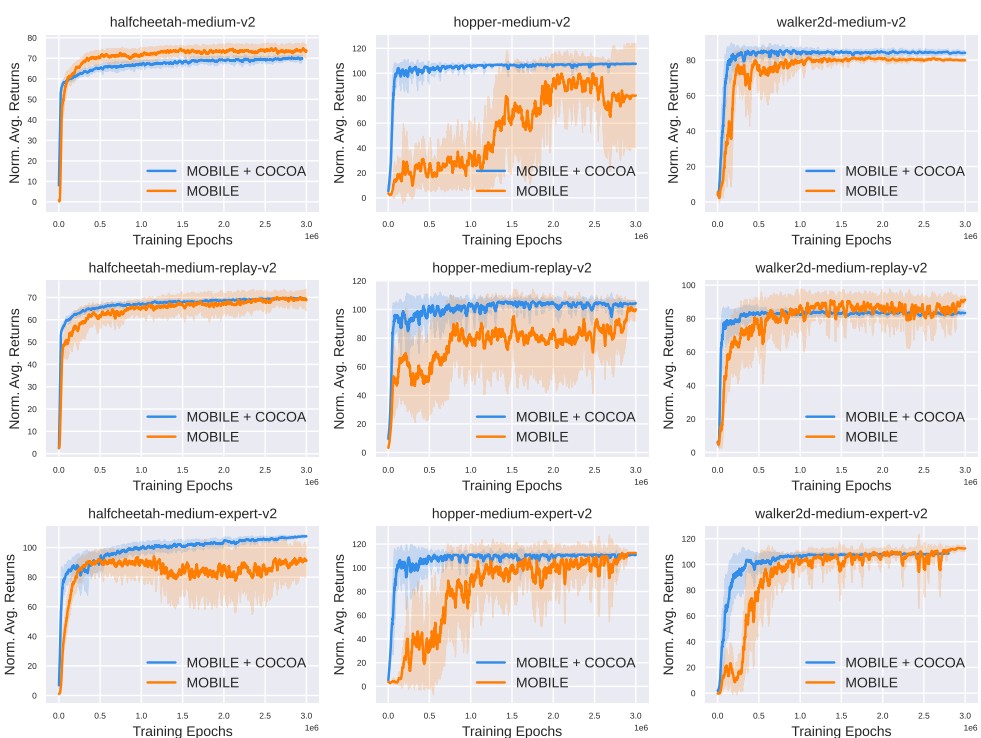

Figure 6: Performance comparison of MOBILE and MOBILE+COCOA across all D4RL tasks except for "random" tasks.

