# OpenReview forum: "Compositional Conservatism: A Transductive Approach in Offline Reinforcement Learning"
_ICLR.cc/2024/Conference — ICLR 2024 poster_

### Official Review · Reviewer_7X6Q · 2023-10-24

**Soundness:** 3 good
**Presentation:** 3 good
**Contribution:** 2 fair
**Rating:** 5
**Confidence:** 3

**Summary:**

Authors propose method COCOA for adapting bilinear transduction to offline RL setup with training environment dynamics and anchor-seeking policy.

**Strengths:**

Proposed approach may boost the performance of the offline RL algorithms and applicable to any actor-critic algorithm. Approach is tested on 4 different algorithms.

**Weaknesses:**

* Training world models might be very time-consuming.

* Proposed approach lead to the improvement only in approximately 58% of cases. And performance might be decreased dramatically. I've also calculated the differences between "+ COCOA" and "Alone" in Table 1 and it appeared that COCOA decreases performance down by 0.8 points. This indicates that only specific algorithms are expected to benefit on average by COCOA. And for MOPO and MOBILE hyperparameters were heavily tuned for each dataset which might be the cause why they benefited.

* Evaluation is performed only on the Gym MuJoCo datasets which I think is not enough now and evaluation on D4RL AntMaze or Adroit is essential. As an alternative, offline-to-online setup might be tested.

**Questions:**

* What is the training time for algorithms with and without COCOA?

* What is hyperparameters sensitivity for MOPO and MOBILE when COCOA is added compared to the same algorithms without COCOA? EOP (https://arxiv.org/abs/2110.04156) can be used for this purpose.

* What are IQL scores on random datasets?

* Are there any thoughts why IQL suffer from COCOA?

* How would COCOA behave with offline RL algorithms which regularize policy? E.g. ReBRAC (https://arxiv.org/pdf/2305.09836.pdf) which is the state-of-the-art algorithm from this family.

* Could you please run experiments on AntMaze or Adroit domains or test your approach in offline-to-online setup? At least for IQL and MOBILE.

* Please add the average scores for each of the approaches in the Table 1 so it is clear what is actual impact of COCOA on algorithms?

Minor text issues: the second paragraph of section 2.1 duplicates information from the first one. Results section has  "**FFor** IQL,".

---

> ### Author Response · Authors · 2023-11-23
> **Authors’ response to Reviewer 7X6Q**
>
> We appreciate the reviewer’s thoughtful and constructive feedback. We revised multiple text issues that the reviewer pointed out. Due to space limits, tables were deferred to our response to Reviewer HX7o or can be found in the paper.
>
> ## Training time for world model and for algorithms with and without COCOA
> Learning dynamics is not a significant burden compared to policy training. The time taken to train dynamics models for each algorithm is about 5-10 percent of the policy training time.
> For IQL as an example, it takes about 1-2 hours on training dynamics, just a few minutes for anchor seeker training and 13 hours for policy training on TITAN RTX GPU. The training time per epoch increased from 11 seconds to 24 seconds as of now.
>
> We mainly focused on the performance improvement that COCOA can bring to existing offline RL algorithms rather than optimizing the running time of our implementation. The majority of the additional training time by COCOA involves frequent calls to the pretrained anchor seeker policy, the dynamics model and the main policy. We expect that training time with COCOA can be greatly reduced through the operator fusion using a toolbox like TorchInductor.
>
> ## Overall performance with COCOA
> Upon investigation following the reviewer's suggestion, we present our updated experimental results with the layer normalization for the anchor seeking policy and the transductive predictor, which turned out to be the missing component for stable training of COCOA. The results are shown in Table 1 in the paper. Comparison with reproduced IQL performance on Gym-MuJoCo random tasks are also included in Appendix. Note that we add the layer normalization only to the COCOA-specific components for a fair comparison with the baselines. As a result, COCOA increases the average performance of all base algorithms evaluated on the D4RL Gym-MuJoCo benchmark. The winning task ratio also becomes 63%. This suggests that with a proper training stabilization, adding COCOA can be beneficial across diverse base algorithms.
>
> ## Other benchmarks for COCOA
> Following the reviewer’s suggestion, we experiment with two more benchmarks - D4RL Adroit and NeoRL [5]. The outcomes of these experiments are summarized in Table 2, 3 in our response to Reviewer HX7o.
> This broader analysis reveals that COCOA significantly enhances the performance of IQL and MOBILE in most tasks. For a detailed insight into the experimental setup, please refer to the updated paper.
>
> Our method consistently enhances the performance across six tested D4RL Adroit tasks, showcasing its robustness and adaptability. In the difficult tasks like the door and hammer due to sparse rewards, the baselines work poorly, and our COCOA can still improve them. On the other hand, in the pen task, our method achieves significant performance improvements.
>
>
>
>
> We also verify the effectiveness of our algorithm in the NeoRL benchmark, known for its conservative, less exploratory data generation compared to the D4RL Gym-MuJoCo. Although its state-action space coverage is highly limited, our preliminary experiments, conducted using MOPO, show significant performance improvement, suggesting potential benefits for other algorithms as well, which will be done for the camera-ready.
>
> Due to the time constraint of the rebuttal period, we test COCOA on six D4RL Adroit tasks on top of IQL and MOBILE, with plans to expand to CQL, MOPO, and ReBRAC in the camera-ready, and initially tested only with MOPO for NeoRL, intending to examine more algorithms later.
>
> ## Comparison with ReBRAC
> Comparing with offline RL methods that use policy regularization would make our results more comprehensive. However, due to the limited time frame, our work on incorporating ReBRAC with COCOA into a single framework is still in progress. We are hoping to include the results in the camera-ready version.
>
> ## Hyperparameter sensitivity
> Regarding MOPO and MOBILE's hyperparameter sensitivity, due to the time and resource constraints, we could not perform exhaustive analysis like EOP. We found that the `rollout length` and `penalty coefficient` need to be slightly larger than those used in the existing MOPO and MOBILE methods. While MOPO and MOBILE conduct the searches for `rollout length` and `penalty coefficient` with 2 and 4, or 2 and 5 values respectively, we use our search with 3 and 5 values.
>
> We appreciate the reviewer’s time and effort put into improving our paper.
>
> [1] Kumar, Aviral, et al. (2020). Conservative q-learning for offline reinforcement learning. NeurIPS.
> [2] Kostrikov, Ilya, et al. (2022). Offline Reinforcement Learning with Implicit Q-Learning. ICLR.
> [3] Yu, Tianhe, et al. (2020). Mopo: Model-based offline policy optimization. NeurIPS.
> [4] Sun, Yihao, et al. (2023). Model-Bellman Inconsistency for Model-based Offline Reinforcement Learning. ICML.
> [5] Rongjun Qin, et al. (2022). NeoRL: A Near Real-World Benchmark for Offline Reinforcement Learning. NeurIPS

---

### Official Review · Reviewer_vdDy · 2023-10-31

**Soundness:** 3 good
**Presentation:** 3 good
**Contribution:** 3 good
**Rating:** 8
**Confidence:** 3

**Summary:**

This paper proposes a new form of conservatism in offline RL called COCOA, specifically with "anchor states" defined by a learned dynamics model. The use of anchor states is motivated by work transforming the out-of-support generalization problem to an out-of-combination problem in the offline RL context.

Experiments show that the addition of COCOA to common offline RL algorithms results in improved performance across the D4RL benchmark suite, and ablations show that the proposed anchor-seeking method is crucial for good performance across said domains.

**Strengths:**

Overall, the paper is very well written, and generally easy to follow. While I haven't worked in this area in RL research a lot, it is interesting to see the extension of a supervised learning approach for use in offline RL, and for it to work as well as it does seem to work here.

The performance results are very interesting, and show that anchor-seeking methods can improve performance especially in the medium-expert domains, where maybe excess conservatism leads to suboptimal performance for most offline RL algorithms. The experiments and ablations are generally solid and thorough, across many relevant benchmarks.

**Weaknesses:**

It seems that this model-based approach here is similar to those in forward model-based offline RL algorithms such as COMBO [1], where the goal is to use a larger coverage state-action distribution for offline RL through the use of a forward dynamics model. It would be interesting to see the comparison between this approach (combined with CQL) and COMBO (if COMBO results can be reproduced), as COMBO also shows significantly less conservatism when used with CQL compared to CQL on its own. I wouldn't say that this is a big weakness, but I feel like from a high level it would be worth it to do this comparison.

There has also been work showing that even synthetic experience replay [2] from strong generative models is useful for RL -- might be worth citing or comparing against as well, but I may be wrong here. Again, not a huge weakness in my eyes but would be interesting to see the comparison.

[1] Tianhe Yu, Aviral Kumar, Rafael Rafailov, Aravind Rajeswaran, Sergey Levine, Chelsea Finn; COMBO: Conservative Offline Model-based Policy Optimization (NeurIPS 2021)

[2] Cong Lu, Philip J. Ball, Yee Whye Teh, Jack Parker-Holder; Synthetic Experience Replay (NeurIPS 2023)

**Questions:**

No real big questions here actually -- very solid!

**Details Of Ethics Concerns:**

None.

---

> ### Author Response · Authors · 2023-11-23
> **Authors’ response to Reviewer vdDy**
>
> We appreciate the reviewer’s thoughtful and constructive feedback.
>
> ## Comparison with COMBO algorithm
> CQL+COCOA and COMBO exhibit some similarities, notably in their use of dynamics and a less conservative approach towards state-action space. However, their methodologies in pursuing conservatism differ significantly: COCOA focuses on conservatism in the compositional input space, whereas COMBO emphasizes regularizing the values for unfamiliar actions. Thus, COCOA and COMBO are orthogonal, it would be an insightful comparison to integrate COCOA with COMBO, as COCOA can be an add-on to any algorithm.
>
> Similar to COMBO, MBPO-based methods like MOPO and RAMBO also show a tendency to outperform model-free methods in random and medium settings [1,2]. It seems that data augmentation through MBPO is particularly beneficial in these tasks. It would be interesting to theoretically or empirically compare the state-specific value functions among CQL, CQL+COCOA, and COMBO for further analysis.
>
> Table 4: Comparison of CQL+COCOA and COMBO on D4RL Gym-MuJoCo benchmark.
> | Task | CQL | CQL+COCOA | COMBO |
> |----------------------------|-------|-----------|-------|
> | halfcheetah-random | 31.3 | 23.8 | 38.8 |
> | hopper-random | 5.3 | 8.8 | 17.9 |
> | walker2d-random | 5.4 | 5.5 | 7 |
> | halfcheetah-medium | 46.9 | 49 | 54.2 |
> | hopper-medium | 61.9 | 66 | 97.2 |
> | walker2d-medium | 79.5 | 83.1 | 81.9 |
> | halfcheetah-medium-replay | 45.3 | 46.4 | 55.1 |
> | hopper-medium-replay | 86.3 | 96.5 | 89.5 |
> | walker2d-medium-replay | 76.8 | 83.9 | 56 |
> | halfcheetah-medium-expert | 95 | 92.4 | 90 |
> | hopper-medium-expert | 96.9 | 103.3 | 111.1 |
> | walker2d-medium-expert | 109.1 | 109.4 | 103.3 |
> | **Average** | **61.64** | **64.01** | **66.83** |
>
>
> ## Comparison with synthetic experience replay (SynthER)
>
> Thank you for proposing an interesting perspective on the utilization of the dataset. The dataset augmentation using dynamic models could be substituted with that using synthetic experience replay. While the selection of the augmentation strategy is orthogonal to COCOA. It seems extremely interesting to investigate how the synthetic experience replay with a generative model synergizes with our algorithm.
>
> We appreciate the reviewer’s time and effort put into improving our paper.
>
> [1] Yu, Tianhe, et al. (2020). Mopo: Model-based offline policy optimization. NeurIPS.
> [2] Sun, Yihao, et al. (2023). Model-Bellman Inconsistency for Model-based Offline Reinforcement Learning. ICML.
> [3] Marc Rigter, Bruno Lacerda, & Nick Hawes (2022). RAMBO-RL: Robust Adversarial Model-Based Offline Reinforcement Learning. NeurIPS.

---

### Official Review · Reviewer_HX7o · 2023-11-02

**Soundness:** 3 good
**Presentation:** 3 good
**Contribution:** 3 good
**Rating:** 6
**Confidence:** 2

**Summary:**

The paper presents an approach to enhance offline reinforcement learning by reparameterizing out-of-distribution states into two simpler components: an anchor and a difference from the anchor, leveraging bilinear transduction for better generalization. The proposed anchor-seeking policy, which utilizes reverse model rollouts, aims to identify these components from within the distribution seen during training, ensuring relevance and improving computational efficiency. The policy is complemented by a reverse dynamics model that diversifies the training data, allowing the model to generalize to new environments effectively. This novel method has been empirically shown to enhance the performance of several state-of-the-art offline RL algorithms.

**Strengths:**

Pros:
1. The paper is well-written and easy to follow.  The reproducibility is good as it provided code.
2. Introduce a novel perspective in finding conservatism in the compositional input space, different from the previous works in finding conservatism in the behavioral space.
3. COCOA can be used in-combination with other offline RL algorithms to improve the performance of previous model-free and model-based algorithms.
4. Good experimental results with SOTA performance on the D4RL benchmarks.
5. Ablation study in comparing with a baseline anchor selection process demonstrate that their anchor-seeking methods is crucial for their performance.

**Weaknesses:**

Cons:
1. Why not train a reverse policy but use a random divergent reverse policy? Is there an ablation study on this?
2. The evaluation is only limited to D4rl benchmarks and why in IQL, adding COCOA does not improve much among all the d4rl tasks, could you provide some insights into this?

**Questions:**

please see weakness

---

> ### Author Response · Authors · 2023-11-23
> **Authors’ response to Reviewer HX7o**
>
> We appreciate the reviewer’s thoughtful and constructive feedback.
>
> ## Choice of the reverse policy
> An effective reverse rollout policy should diverge from an offline dataset and encompass various terminal states. We avoided training a reverse policy because defining a precise loss function for it would involve quantifying (1) the state's distance from the dataset distribution and (2) the terminal state diversity, which is computationally intensive.
>
> Hence, we employed a random divergent reverse policy while comparing COCOA with trained reverse policy as an ablation study. Similar to [1], we employed a state-conditioned conditional variational autoencoder[2] that mimics the offline dataset's behavior in reverse. The performance of MOPO+COCOA on two D4RL Gym-MuJoCo tasks showed that a random diverse policy is superior for high performance.
>
> Table 1:  Ablation study on the reverse policy.
> | Task | Trained | Random |
> |----------------------------|------------|----------|
> | halfcheetah-medium-replay | 58.8 | 72.1 |
> | walker2d-medium-replay | 68.6 | 92.3 |
>
> ## Other benchmarks for COCOA
>
> Following the reviewer’s suggestion, we experimented with two more benchmarks - D4RL Adroit and NeoRL [3]. The outcomes of these experiments are summarized in the tables below.
> This broader analysis reveals that COCOA enhances the performance of IQL and MOBILE in most tasks. For a detail about experimental setup, refer to the updated paper.
>
> Our method demonstrated consistent performance enhancements across six D4RL Adroit tasks we tested, showcasing its robustness and adaptability. While COCOA encountered challenges in complex tasks like the door and hammer, akin to its original algorithm, this reflects the inherent difficulty of these tasks due to sparse rewards. Notably, in tasks like the pen, our method achieved noticeable performance improvements.
>
> We also verified the effectiveness of our algorithm in the NeoRL benchmark, which is known to be more difficult than the traditional D4RL Gym-MuJoCo benchmark as it generates the data in a more conservative and less exploratory manner. The data collected in this manner have less coverage in the state-action space, posing the need for more fine-grained stitching. In this perspective, we can see that the dataset's challenges align with our algorithm.
> To check whether our algorithm also performs well in NeoRL, we firstly conducted experiments with MOPO which demonstrates an impressive performance improvement. Based on these enhancements, we anticipate that other algorithms would also have a big advantage with our algorithm.
> Under time constraints, we tested COCOA on six D4RL Adroit tasks on top of IQL and MOBILE, with plans to expand to CQL, MOPO, and ReBRAC in the final paper, and initially tested only with MOPO for NeoRL, intending to examine more algorithms later.
>
> Table 2: D4RL Adroit benchmark results.
> | Task | IQL | IQL+COCOA | MOBILE | MOBILE+COCOA |
> |------------------|--------|-----------|---------|--------------|
> | door-cloned-v1 | 2.11 | 2.81 | -0.27 | 0.92 |
> | door-human-v1 | 5.27 | 6.81 | -0.28 | -0.05 |
> | hammer-cloned-v1 | 0.53 | 0.57 | 0.23 | 0.29 |
> | hammer-human-v1 | 1.33 | 2.21 | 0.25 | 1.11 |
> | pen-cloned-v1 | 72.55 | 73.98 | 54.95 | 48.9 |
> | pen-human-v1 | 74.4 | 78.15 | 14.76 | 41.65 |
> | **Average** | **26.03** | **27.22** | **11.61** | **15.47** |
>
>
> Table 3: NeoRL benchmark results.
> | Task | MOPO | MOPO+COCOA |
> |-----------------|-------|------------|
> | HalfCheetah-L | 40.1 | 47.47 |
> | Hopper-L | 6.2 | 23.02 |
> | Walker2d-L | 11.6 | 14.23 |
> | HalfCheetah-M | 62.3 | 78.26 |
> | Hopper-M | 1 | 61.65 |
> | Walker2d-M | 39.9 | 49.80 |
> | HalfCheetah-H | 65.9 | 42.44 |
> | Hopper-H | 11.5 | 29.31 |
> | Walker2d-H | 18 | 53.38 |
> | **Average** | **28.5** | **44.40** |
>
> ## The performance of IQL
>
> Especially in the context of IQL, we observed that certain tasks occasionally exhibited a sudden drop in performance. Based on the guidance you provided, and considering that model-free algorithms are typically more challenging to train compared to model-based algorithms (as they do not augment the dataset with a dynamics model), we finalized our network design to employ the layer normalization technique. This approach is particularly much more effective in tasks especially such as 'hopper-medium-replay', 'walker 2d-medium-replay' and so forth.
>
> We appreciate the reviewer’s time and effort put into improving our paper.
>
> [1] Wang, Jianhao, et al. (2021). Offline Reinforcement Learning with Reverse Model-Based Imagination. NeurIPS.
> [2] Sohn, Kihyuk, et al. (2015). Learning structured output representation using deep conditional generative models. NeurIPS.
> [3] Rongjun Qin, et al. (2022). NeoRL: A Near Real-World Benchmark for Offline Reinforcement Learning. NeurIPS

---

### Meta-Review · Area_Chair_f7Ld · 2023-12-07

**Metareview:**

The paper presents an approach to enhance offline reinforcement learning by reparameterizing out-of-distribution states into two simpler components: an anchor and a difference from the anchor, leveraging bilinear transduction for better generalization. The proposed anchor-seeking policy, which utilizes reverse model rollouts, aims to identify these components from within the distribution seen during training, ensuring relevance and improving computational efficiency. The policy is complemented by a reverse dynamics model that diversifies the training data, allowing the model to generalize to new environments effectively. This novel method has been empirically shown to enhance the performance of several state-of-the-art offline RL algorithms.

The approach is quite interesting and novel compared with the existing approach. While the experimental results are not particularly impressive, AC believes there is potential to generate further insights to the community. Therefore, the AC recommends acceptance.

**Justification For Why Not Higher Score:**

The experimental results are not particularly impressive as the proposed method does not dominate other approaches in all cases.

**Justification For Why Not Lower Score:**

The proposed method is interesting and may generate further insights to the community.

---

### Decision · Program_Chairs · 2024-01-16

Accept (poster)